# Ex Vivo Generation and Characterization of Human Hyaline and Elastic Cartilaginous Microtissues for Tissue Engineering Applications

**DOI:** 10.3390/biomedicines9030292

**Published:** 2021-03-12

**Authors:** David Sánchez-Porras, Daniel Durand-Herrera, Ana B. Paes, Jesús Chato-Astrain, Rik Verplancke, Jan Vanfleteren, José Darío Sánchez-López, Óscar Darío García-García, Fernando Campos, Víctor Carriel

**Affiliations:** 1Department of Histology, Tissue Engineering Group, Faculty of Medicine, University of Granada, 18016 Granada, Spain; davidsp@go.ugr.es (D.S.-P.); durandherrera@gmail.com (D.D.-H.); 2Instituto de Investigación Biosanitaria ibs. GRANADA, 18012 Granada, Spain; jchato@go.ugr.es (J.C.-A.); e.oscargg@go.ugr.es (Ó.D.G.-G.); 3Doctoral Program in Biomedicine, Doctoral School, University of Granada, 18016 Granada, Spain; 4Master Program in Tissue Engineering and Advanced Therapies, International School for Postgraduate Studies, University of Granada, 18016 Granada, Spain; anapaesmarti@gmail.com; 5Centre for Microsystems Technology (CMST), imec and Ghent University, 9052 Ghent, Belgium; rik.verplancke@ugent.be (R.V.); jan.vanfleteren@ugent.be (J.V.); 6Division of Maxillofacial Surgery, University Hospital Complex of Granada, 18013 Granada, Spain; josed.sanchez.sspa@juntadeandalucia.es

**Keywords:** microtissues, tissue engineering, human hyaline chondrocytes, human elastic chondrocytes, extracellular matrix, organoids

## Abstract

Considering the high prevalence of cartilage-associated pathologies, low self-repair capacity and limitations of current repair techniques, tissue engineering (TE) strategies have emerged as a promising alternative in this field. Three-dimensional culture techniques have gained attention in recent years, showing their ability to provide the most biomimetic environment for the cells under culture conditions, enabling the cells to fabricate natural, 3D functional microtissues (MTs). In this sense, the aim of this study was to generate, characterize and compare scaffold-free human hyaline and elastic cartilage-derived MTs (HC-MTs and EC-MTs, respectively) under expansion (EM) and chondrogenic media (CM). MTs were generated by using agarose microchips and evaluated ex vivo for 28 days. The MTs generated were subjected to morphometric assessment and cell viability, metabolic activity and histological analyses. Results suggest that the use of CM improves the biomimicry of the MTs obtained in terms of morphology, viability and extracellular matrix (ECM) synthesis with respect to the use of EM. Moreover, the overall results indicate a faster and more sensitive response of the EC-derived cells to the use of CM as compared to HC chondrocytes. Finally, future preclinical in vivo studies are still needed to determine the potential clinical usefulness of these novel advanced therapy products.

## 1. Introduction

Tissue engineering (TE) is an interdisciplinary field that applies the principles of engineering and life sciences toward to the development of bioengineered substitutes to restore or maintain the structure and function of tissues or organs [1,2]. To engineer living tissues in vitro, cultured cells are induced to grow within different kinds of scaffolds, or they are stimulated to create their own extracellular matrix (ECM). Both methods allow generating 3D bioartificial substitutes, with controlled biomechanical, structural and biological properties, to promote tissue regeneration in vivo [3,4,5].

Cartilage is an avascular, aneural and alymphatic tissue composed of one cell population immersed in a highly specialized ECM. These cells are the chondrocytes, which in the immature state (chondroblasts) have a high metabolic activity and are responsible for the ECM synthesis and organization. Concerning the ECM, in general, cartilages have a complex collagen fiber network—in specific cases elastic fibers—and a high amount of non-fibrillar ECM molecules, especially sulfated proteoglycans [6]. This ECM provides a high hydration rate, structural stability, flexibility and adaptive properties (compression and deformation) during function [7]. Based on the molecular composition of the ECM, in humans, we can find three varieties of cartilaginous tissues, the hyaline, the elastic and the fibrous cartilage [6]. In addition, among the collagens that can be found in all cartilage ECMs, the type II collagen is the most common, whereas the type I collagen is mainly found in fibrous cartilage and scarcely observed in the other types [6,8,9]. However, the elastic fibers are exclusively found in elastic cartilage which is related to the biomechanics of these tissues [10].

Clinically, hyaline and elastic cartilages are often affected by different kinds of pathological conditions and/or traumatic injuries. Unfortunately, due to the limited ability of cartilage to self-repair, injuries usually remain for years and can eventually lead to further degeneration and dysfunction [11,12]. For this reason, science has tried for decades to find a way to repair or regenerate damaged or lost hyaline and elastic cartilages [9,12,13,14,15,16], but their complex structure and ECM composition resulted in being difficult to recreate [11,12,17,18]. Therefore, cartilage became a potential candidate for TE and cell-based therapies. In this context, the generation of 3D microtissues (MTs) resulted in a promising alternative in TE and cartilage [19,20,21,22,23,24,25].

The generation of 3D MTs offers the possibility to create a controlled microenvironment, ex vivo, that could resemble some structural, molecular and physiological conditions of native tissues, avoiding the use of biomaterials [4,26,27,28]. These MTs can be generated with diverse techniques, which in general use non-adherent molds or microwells [26,29]. Depending on the geometry of the molds or microwells used, these techniques allow defining the MT size and dimensions, and the 3D architecture (rods, spheroids and 3D patterns) where one or even more cell types can be included [26,30,31,32]. From the cellular point of view, this technique forces the cell–cell interactions, thus stimulating the synthesis of diverse and tissue-specific ECM molecules, resulting in the formation of 3D and well-defined MTs. Similar methodologies were used to generate neurospheres [33], skin-derived fibroblasts [34], epithelial-based organotypic 3D spheres [35] and chondrogenic ones [22,23]. However, in the case of cartilage, a wide range of methodologies and culture conditions have been used with highly variable and heterogeneous results. Therefore, research is still needed to determine the behavior and the essential biological properties of the chondrocyte-derived MTs for TE applications.

In this sense, the aim of this study was to generate, characterize and compare human hyaline and elastic cartilage-derived MTs or organoids (HC-MTs and EC-MTs, respectively), under chondrogenic and expansion culture conditions. Furthermore, generated MTs were subjected to morphometry, cell viability, cell metabolic activity and ECM synthesis analyses during 28 days of ex vivo development (EVD).

## 2. Experimental Section

### 2.1. Cell Cultures

Chondrocytes were isolated from hyaline and elastic human cartilage biopsies from healthy donors under informed consent and processed following previously described protocols [10]. Briefly, biopsies were fragmented in small pieces and digested in a collagenase I solution at 37 °C. The cells were harvested by centrifugation and then cultivated using EM consisting of Dulbecco’s modified Eagle’s medium (DMEM; Sigma-Aldrich, Munich, Germany)/nutrient mixture F12 (F12) supplemented with 10% fetal bovine serum (FBS), 1% antibiotic solution and 1% L–glutamine and kept under standard culture conditions (37 °C and 5% CO_2_) [10]. Culture media were renovated every 3 days, and confluency was daily controlled by phase-contrast microscopy. Subconfluent cells were detached with trypsin-EDTA and subcultured in new culture flasks at 10^4^ cells/cm^2^. Cells were expanded until the fifth passage, which allowed obtaining enough number of cells for MT generation. All products were purchased from Sigma-Aldrich.

This study was approved by the local ethical committee of the University of Granada (Grant nº CS PI-0257-2017).

### 2.2. Fabrication of Molds

Micro-patterned elastomeric stamps were fabricated using replica molding. Master molds were prepared by spin coating a 200-µm-thick uniform layer of SU-8 100, i.e., a negative tone, epoxy-based photoresist developed by MicroChem, on a silicon wafer. The SU-8 was then selectively developed after exposing it to UV (high-pressure mercury vapor lamp, central wavelength 365 nm, power 13 mW/cm^2^, duration 50.7 s) through a transparent photomask containing black circular dots with radii of 400 µm, yielding cylindrical wells in the SU-8 with a depth of 200 µm and radii identical to those of the circular dots. After hard baking the master mold for 90 min at 120 °C in a convection oven, polydimethylsiloxane (PDMS) was prepared by mixing Sylgard^®^ 184 (Dow Corning, Midland, MI, USA) base and curing agent in a 10:1 ratio by weight, respectively. The liquid prepolymer was then poured on the master mold until a thickness of approximately 2 mm was achieved. After degassing, it was left to cure for 48 h at room temperature, thereby replicating the surface topography of the master mold. The cured elastomer was subsequently peeled from the master mold and circular discs with a diameter of 18 mm were manually punched out of the PDMS.

### 2.3. Fabrication of Agarose Microchips

A 3.5% agarose solution was prepared by dissolving Type I agarose in sterile phosphate-buffered saline (PBS) (both from Sigma-Aldrich). PDMS molds were used as a negative replica of the microwells, and the PDMS mold contained 1041 microwells with a flat bottom, a diameter of 400 µm and a depth of 200 µm. The molds were first sterilized in 70% ethanol and then placed in a 12-well plate, where the agarose solution was poured. The agarose was placed at 4 °C for 1 h and then removed and finally transferred to a sterile 12-well suspension plate, rinsed in PBS and stored at 4 °C until use [4].

### 2.4. Formation of Chondrogenic Microtissues or Organoids

First, the microchips were equilibrated for 30 min with 2 mL of DMEM/F12 followed by the seeding of 2.5 × 10^5^ hyaline or elastic cartilage-derived chondrocytes in 0.5 mL of culture media. In this study, cells were equally divided into the following groups:(a)HC-MT cultured under EM as control (HC-MT-EM);(b)HC-MT cultured with CM (HC-MT-CM);(c)EC-MT cultured with EM as control (EC-MT-EM);(d)EC-MT cultured with CM (EC-MT-CM).

Plates were placed to rest for 1 h to let the cells settle by gravity at the bottom of the microwells. The EM was described above, while the CM was composed of DMEM/F12 supplemented with 1% antibiotic solution, 5 mg/mL insulin transferrin-selenite (Sigma-Aldrich), 100 nM dexamethasone (Sigma-Aldrich), 10 ng/mL TGF-β (Sigma-Aldrich), 100 µg/mL sodium pyruvate (Gibco, Grand Island, NY, USA), 200 µM L-ascorbic acid 2-phosphate (Sigma-Aldrich) and 0.35 µM L-proline (Sigma-Aldrich) [36]. Finally, once settled, 1.5 mL of EM or CM was added, and microchips were kept under standard culture conditions (37 °C and 5% CO_2_). In addition, HC and EC cells were cultured in standard 2D conditions with EM or CM as control.

MTs were kept under culture conditions with EM or CM during 28 days of EVD concretely at 4, 7, 14, 21 and 28 days. MTs were harvested from 4 days of EVD onward and subjected to different analyses listed below. An exception was made in the case of the morphometric analyses.

### 2.5. Morphometric Analyses

Phase-contrast microscopy was routinely used to evaluate the chondrocyte MT formation process in the microchips (Nikon, Eclipse Ti-U, Tokyo, Japan). Microscopic images were taken at different time points (0, 1, 2, 3, 4, 7, 14, 21 and 28 days of EVD), from each experimental group, and then subjected to morphometric analysis with the NIS Elements Advanced Research Nikon software. In this context, the area, diameter, perimeter, volume and circularity were determined in 10 MTs from each experimental condition following a recently described procedure [4]. Circularity was calculated using the formula: Circularity = (4πA)/p^2^, considering that a perfect circle has a value of 1.

### 2.6. Assessment of the Cell Viability and Metabolic Activity during MT Formation

In this study, the MTs generated were subjected to cell viability, irreversible cell membrane damage and cellular metabolic activity characterization using Live/Dead Cell Viability Assay (Invitrogen, Carlsbad, CA, USA), DNA-release quantification and WST-1 assays (Sigma-Aldrich), respectively, following the manufacturers’ recommendations and previously described protocols [4,37]. Tests were performed in three independent microchips at 4, 7, 14, 21 and 28 days of EVD. To perform Live/Dead assay, microchips were washed with PBS and MTs were incubated with calcein AM/ethidium homodimer-1 (Invitrogen) for 10 min. Viability of the MTs was evaluated by fluorescence microscopy (Nikon, Eclipse Ti-U inverted research system microscope, NIS Elements Advanced Research software) [20,38,39]. It was not possible to perform quantitative analyses of viable and dead cells due to the tridimensional structure of the MTs. Released DNA was quantified to determine the irreversible cell damage, following previously described methods [40]. The culture media were collected and quantified by spectrophotometry by using a NanoDrop 2000 Spectrophotometer (Thermo Fisher Scientific, Waltham, MA, USA) [4]. To evaluate cell metabolic activity of the MT-forming cells, the MTs were incubated with the WST-1 reagent for 4 h at 37 °C following the manufacturer’s recommendations and previously described protocols [37,39,41]. The colorimetric reaction was measured with an ASYS UVM340 spectrophotometer and the software DigiRead (Biochrom Ltd., Cambridge, UK). Within all tests, 2D cultures of each cell type were used as positive controls. As negative controls, 2D cultures were treated with a 1% Triton X-100 solution, which induces an irreversible cell membrane damage as described previously [4,37]. In addition, all tests were performed in triplicate.

### 2.7. Histological Analyses and Histochemical Analysis

For these analyses, three independent microchips with MTs were harvested and fixed in 3.7% neutral buffered formaldehyde, dehydrated and embedded in paraffin, as previously described in [42]. Tests were performed on three independent microchips at 4, 7, 14, 21 and 28 days of EVD.

To evaluate cell morphology and MT histoarchitecture, sections were stained with hematoxylin–eosin (HE) using a standard protocol. Acidic proteoglycans were identified with the Alcian blue (AB) histochemical method at pH 2.5, while the fibrillar collagen fibers were determined with Picrosirius staining (PS). The elastic fibers were evaluated with orcein histochemical staining.

Immunohistochemistry (IHC) was used to determine the chondrogenic profile through the identification of collagen type II (Rabbit polyclonal, EMD Millipore, CN:AB2036), collagen type IV (Mouse monoclonal (clone CIV22+PHM-12), Master Diagnóstica, CN: MAD001060QD) and the S-100 protein (Rabbit polyclonal DakoCytomation, CN: Z0311). Furthermore, PCNA (Mouse monoclonal (clone PC10), Sigma-Aldrich, CN: P8825) immunodetection was used to identify proliferating cells. All these procedures were conducted as previously described [4,10,43,44]. Moreover, IHC stainings were performed at the same time and under the same technical conditions. As negative technical controls, the incubation of the primary antibody was omitted in a separate slide for each antibody used.

### 2.8. Statistical Analysis

For morphometric parameters, cellular metabolic activity and DNA values, the Mann–Whitney nonparametric test was used to determine significant differences among groups. Correlations between morphometric parameter values were assessed by the Kendall nonparametric test. Significant differences were considered for *p* < 0.05, and all results are shown as mean ± standard deviation (SD) values. All statistical comparisons were performed with the software SPSS 24.0.

## 3. Results

### 3.1. Microtissue Formation and Morphometric Analysis

After seeding, both cell types under EM or CM were found concentrated at the bottom of the microwells and progressively formed spheroid MTs from 24 h onward. In general, the characteristic spheroid structure of the MTs was obtained and it was relatively stable over the time, especially when CM was used (Figure 1). Curiously, in the case of the EC-MT-EM group, the MTs were poorly defined and characterized by irregular edges with some signs of disaggregation, especially from 7 to 28 days of EVD (Figure 1). Moreover, from 14 days onward, an irregular material (cellular debris) around EC-MT-EM started to be produced, being more evident at 28 days of EVD. Furthermore, microscopic evaluation showed differences concerning the circularity and MTs’ size, which were accurately confirmed by the quantitative analyses (Figure 1 and Figure 2).

Quantitative analyses showed that the progressive formation of MTs by using HC and EC under EM and CM resulted in the development of relatively stable and circular 3D structures during the 28 days of EVD (Figure 1 and Figure 2). Concerning the area of the MTs generated, the average values for HC under EM and CM were 11.2 × 10^3^ and 11.25 × 10^3^ µm^2^, respectively, without significant differences (*p* = 0.703, Figure 2). Once the MT was established at 24 h, the area of HC under EM and CM experienced a progressive reduction or compaction over time. This progressive MT formation process was accompanied by an increase in the MT’s circularity, and proportional changes in other morphological parameters (diameter, perimeter and volume), with some statistical differences (see values and significant differences in Figure 2). Furthermore, statistical analyses revealed that HC-MT-CM was consistently more circular than HC-MT-EM (*p* = 0.001), with specific differences at 4, 7 and 28 days of EVD (*p* < 0.05, Figure 2). In relation to the EC, the MT generated under CM conditions showed relatively comparable behavior to the HC, meaning a reduction in the area and progressive compaction over time (Figure 2). However, in the case of EC-MT-EM, the area showed a comparable behavior, but the overall circularity was significantly lower than EC-MT-CM (*p* = 0.000), and these differences were clearly significant from 1 to 28 days of EVD (*p* < 0.05, Figure 2), confirming the microscopic findings described above. Results corresponding to the diameters and volume are not shown within Figure 2 because they showed exactly the same behavior over time as the area and perimeter values.

When we compared HC-MTs vs. EC-MTs, we observed some differences in morphometry over time and in the function of the culture media used (Figure 2). Concerning overall area values, we did not observe significant differences between HC-MT-EM vs. EC-MT-EM (*p* = 0.400) and HC-MT-CM vs. EC-MT-CM groups (*p* = 0.318), but circularity was significantly higher in HC-MT-EM than EC-MT-EM (*p* = 0.000). Despite these results, the analysis of these parameters in specific periods of time showed some significant differences between HC-MTs vs. EC-MTs, highlighting the structural consistency of HC-MT-EM vs. EC-MT-EM (Figure 2).

The Kendall correlation test showed a positive and significant correlation between the circularity and time of EVD in all experimental conditions (*p* < 0.05). In addition, a positive and significant correlation between the area, diameter, perimeter and volume was observed in all conditions (*p* < 0.05). However, the circularity and dimensions-related parameters were negatively correlated (*p* < 0.05). Finally, the Kendall correlation test showed that with the advance of the EVD days, the MTs suffer a clear compaction process accompanied by an increase in the circularity of the MT generated.

### 3.2. Cell Viability

The Live/Dead assay was performed in both MTs (HC-MTs and EC-MTs) after 4, 7, 14, 21 and 28 days of EVD. Results reveal that both cell type-based MTs generated, under expansion and chondrogenic culture conditions, were mainly composed of viable cells. Interestingly, few dead cells during the whole study were identified, and they were slightly more abundant in EC-MTs than HC-MTs (Figure 3).

The quantification of the DNA released by cells during MT formation or 2D cell culture showed variable values between groups, controls and time (Figure 4). In general, we observed that DNA mean values for each experimental condition (HC-MTs and EC-MTs) and culture medium (EM and CM) were lower than mean values obtained in their respective 2D controls (Figure 4). Analyzing the culture media in the MT condition, MT-CM showed statically significant higher values than MT-EM in HC-MTs, whereas in EC-MTs, values were generally higher in MT-EM than MT-CM (Figure 4). Moreover, 2D control values under EM were statistically significantly higher than 2D controls under CM, HC-2D controls being higher than EC-2D controls in both culture media (Figure 4).

The biochemical analysis of the cell metabolic activity carried out with WST-1 assay showed significant differences for each culture medium used, stating that EM allows higher activity than CM in both cell types, as shown in Figure 4, except in EC-2D control cultures where no significant differences were observed between EM and CM (*p* = 0.081). Comparing the MT conditions with their respective controls, MTs, from both cell sources, showed lower metabolic activity than 2D cell cultures and culture media used (Figure 4). Furthermore, EC showed higher values of metabolic activity in comparison with HC. A different metabolic activity pattern for MTs depending on the culture media can also be observed in Figure 4, showing a significant decrease at the latest culture periods in the EM condition for both cell types, while activity had a more stable behavior over time in CM. Figure 4 shows the statistical results for the MT metabolic activity, which were obtained using the Mann–Whitney test.

### 3.3. Histology of the Microtissue

Histological analyses confirmed the formation of spheroidal MT or organoids with a relatively homogeneous cell distribution in HC-MTs over time. Moreover, HC-MTs were more compact and more circular and had well-defined edges when CM was used (Figure 5). The histological evaluation of EC-MT-EM revealed signs of cellular disaggregation (spheroidal cells), irregular edges and poor compaction over time (Figure 5). Interestingly, the use of CM improved the circularity of the EC-MTs which resulted in being composed of more elongated and flattened chondrocytes differing from the pattern observed in EC-MT-EM (Figure 5). In addition, histology of EC-MT-CM showed the presence of some extracellular spaces between elongated chondrocytes over time (Figure 5).

The analysis of the S-100 protein, a chondrogenic marker, was positive in most of the HC-MT-forming chondrocytes over time, without differences between the use of EM and CM (Figure 6A). In the case of EC-MTs, the S-100 protein was positive in some chondrocytes over time in both culture conditions (EM and CM). However, the positivity of S-100 was less abundant in EC-MTs than in HC-MTs (Figure 6A). The evaluation of cell proliferation during MT formation by PCNA IHC revealed the presence of proliferating chondrocytes in all experimental conditions (Figure 6B). In the case of HC-MTs, no clear differences were observed between the uses of EM and CM, and these results are comparable to the pattern observed in EC-MT-EM. However, a considerable decrease in proliferating chondrocytes was observed within EC-MT-CM between 7 and 28 days of EVD (Figure 6B).

Histochemistry for acid proteoglycans conducted with AB staining revealed that the use of CM induced a progressive and abundant synthesis and extracellular deposition of these molecules in HC- and EC-MTs as compared to the use of EM (Figure 7). The evaluation of fibrillary collagens with PS staining showed some signs of synthesis and extracellular deposition of these fibers in HC-MT-CM and EC-MT-EM, with the PS-positive reaction being less evident in the other groups (Figure 7). In the case of the synthesis and extracellular deposition of collagen type II, it was positive in all experimental groups with a more consistent pattern in HC-MT-CM (Figure 8). When collagen type IV was analyzed, we observed a positive and pericellular pattern in EC-MTs from 7 days of EVD onward. No clear positive reaction was observed in HC-MTs between 4 and 21 days of EVD, with a slight positive reaction after 28 days of EVD without clear differences between the culture media used (Figure 8). In the case of HC-MTs, collagen type IV immunostaining was weak, and a clear reaction was only observed in HC-MT-EM at 28 days of EVD (Figure 8). Finally, the histochemical evaluation of elastic fiber synthesis carried out with orcein histochemical staining did not reveal the presence of these molecules in the MTs generated (data not shown).

## 4. Discussion

Hyaline and elastic cartilaginous tissues can be affected by different pathologies and current therapies have not been able to accomplish the most satisfactory results. In this context, several engineering substitutes have been generated by using different kinds of biofabrication techniques in TE with promising results [45]. In general, biologically active substitutes composed of viable and functional cells combined with natural biomaterials result in better outcomes than the use of synthetic biomaterials [46]. However, it is still a challenge to reproduce the 3D structure, biological activity and ECM molecular composition of cartilaginous tissues by using conventional biofabrication techniques, and therefore new methods which allow obtaining high levels of biomimicry and functionality must be explored.

Currently, it is possible to generate highly biomimetic functional units for TE by using the MT technique. This methodology has been used in different TE applications, demonstrating that cells can progressively produce ECM molecules and establish different kinds of cell–cell or cell–ECM interactions when they are cultured in non-adherent molds, such as agarose microchips [4,31]. It is a relatively easy and inexpensive technique which allows generating large amounts of MTs with predefined and uniform sizes in a controlled manner with high potential for use in cartilage TE [27,47,48]. Attending to the application of MTs in cartilage TE, previous studies demonstrated that chondrocytes can generate, under adequate culture conditions, viable and functional MTs [23]. However, most of these research was mainly conducted with different animal cell sources (such as pigs, cows or rodents) [23,25,49], with these results being difficult to extrapolate to the human cell physiology, tissue microarchitecture and regeneration capability, losing translational potential [50,51].

In this context, special attention has been paid to hyaline cartilage research, whose pathologies are more frequent and prevalent, with the fibrocartilage and especially the elastic cartilage being less studied or characterized [20,23,25,49]. Concerning elastic cartilage TE, researchers are mainly focused on the generation of different kind of substitutes for auricular repair or replacement [10,52,53]. Through the use of the pellet technique, the aggregation and ECM synthesis capability of elastic chondrocytes was demonstrated [54]. Nevertheless, it is still unknown if human elastic cartilage-derived chondrocytes are a suitable cell source for the generation of functional MTs, and the same applies to their biological and structural properties. Additionally, the structural and/or functional similarities or differences between hyaline or elastic cartilage-derived MTs are still unknown, especially when generated under expansion or chondrogenic culture conditions [55]. The knowledge of their features could help to design more functional and biomimetic TE strategies.

Probably two of the most known limitations in cartilage TE are the limited number of cells that can be obtained from a cartilaginous biopsy, and the chondrocyte dedifferentiation that occurs during subculture and expansion ex vivo [56]. These limitations have led researchers to use cartilage grafts from different origins to treat cartilaginous defects with promising, but not optimal, results [9,13]. Actually, it is currently common to use rib hyaline cartilage in auricular surgical reconstruction [57]. More recently, promising results were obtained with the use of a gel containing elastic cartilage-derived chondrocytes in the repair of nasal cartilage [58]. These studies support the hypothesis that hyaline or elastic cartilage-based functional MTs could be potentially used interchangeably in the treatment of diverse cartilage defects. For this reason, here, human hyaline or elastic cartilage-derived chondrocyte MTs were generated by using agarose microchip technology. These MTs were subjected to ex vivo time course (during 28 days of EVD) morphometric, cell viability and functionality and histological analyses.

Concerning the time course phase-contrast microscopy analysis, our results demonstrate that both human cell sources used progressively generate MTs when cultured under EM or CM culture conditions. Nonetheless, clear differences between HC and EC and culture conditions were obtained. The use of CM resulted in the generation of more compact, morphologically stable MTs, which also showed a clear tendency to circularity than those generated with EM, especially when it was applied to the EC. Actually, the EC-MTs generated with EM were more irregular, showed signs of cellular disaggregation and started to deposit cellular debris from 14 days of EVD onward, with these findings being comparable to the behavior of Wharton’s jelly stem cells [4]. This peripheral disaggregation could be related to an increase in integrins within the MTs, due to the fact that there is some evidence which relates to lower MT compaction with expression of these molecules [59]. Therefore, our results demonstrate the impact of the use of CM on cell function and MT stability, with this culture medium being more efficient than the use of EM for the generation of both HC- and EC-based MTs. Our findings are consistent with the degree of compaction and circularity obtained with the use of CM with pig-derived fibrochondrocytes [20], bovine HC and human osteoarthritis-derived chondrocyte micro-aggregates [23].

The impact that MT compaction and homogeneity could have in TE is not yet clear, but there are clear signs of cell–cell and cell–ECM interactions which could make these 3D microstructures more functional and stable. In fact, it was demonstrated that stable MTs are more efficient than the use of cells in suspension in a model of tissue repair and regeneration [48]. Within the MT, diverse cell sources were able to create different kinds of cell–cell and cell–ECM interactions among them, such as N-CAM, cadherins and connexins [47,59,60]. These interactions are related to MT compaction and cell functions, including chondrocytes; moreover, deficiencies on these molecules were found associated with different pathologies [61,62,63]. In light of the results obtained in this study, the use of CM may stimulate the synthesis of some of these molecules, explaining the high degree of compaction and stability obtained. However, additional analyses are still needed to demonstrate the time course evolutive expression of these cell–cell interaction molecules in chondrocyte-based MTs.

MTs generated with CM presented more consistent metabolic activity values during the whole experiment compared with the use of EM. Curiously, the overall metabolic values obtained with all experimental conditions were significantly lower than those observed within the 2D cell culture. This could be explained by the fact that a limited number of MT-forming cells are included in non-adherent microwells, limiting their expansion capability and forcing their association. Interestingly, we found higher metabolic activity in EC-MTs than in HC-MTs, and these findings were correlated with the size (area) of the MTs generated. Concerning the EC-MTs, the use of EM exhibited a decrease in WST-1 values from 14 days of EVD onward, coinciding with the disaggregation signs and DNA peak values. Surprisingly, this behavior was not observed when EC- or HC-MTs were generated with the use of CM, where the consistent morphometric and metabolic profiles were more stable. These differential behaviors between media are in line with previous studies [20,49] where a regular behavior was obtained with the use of CM. The positive impact of CM during HC cell cultures is well known, but this is the first time where the impact of the use of different culture media resulted in a differential behavior between EC- and HC-derived MTs, with these differences being more evident with the EC cell source. Probably the use of CM within MTs favors the quiescent metabolic state of chondrocytes resembling the in vivo chondrocyte physiological behavior [64], explaining the behavior adopted by EC-MTs. However, more molecular studies are needed to elucidate the metabolic pathways involved during these complex processes.

Chondrocytes have a high ECM synthesis capability which progressively decreases with aging [6]. In this context, histological analyses confirmed that both cell sources used to generate MTs were able to progressively produce and deposit diverse ECM molecules over time. Histology confirmed the impact of the use of CM on the MTs’ morphology, cell distribution and ECM synthesis. Indeed, more regular and stable morphology was observed in HC- and EC-derived MTs generated with CM, with these results being in accordance with morphometric and metabolic analyses. Cartilaginous structures are characterized by the molecular composition of the ECM, which also defines the three varieties found in humans [6]. Despite these molecular differences, there are some ECM components which are key for all cartilage varieties, such as collagens (especially type II), proteoglycans and elastic fibers, which are exclusive to ECs. Moreover, chondrocytes also express some common proteins which are used to explore their lineage, such as S-100 which is not chondrocyte-specific [43], but it is widely used for this purpose [10,36]. In this study, MT-forming chondrocytes showed a positive expression of the S-100 protein, confirming their lineage, with these results being more consistent in HC-MTs than those generated by EC chondrocytes. No clear differences were observed in the expression of S-100 based on the use of CM or EM, and thus both cell sources kept their lineage as previously observed in other tissue engineering studies [10,36]. In addition, the immunohistochemical assessment of the PCNA protein confirmed signs of cell proliferation during the whole study in most of the MTs generated. The HC-derived chondrocytes showed consistent cell proliferation over time without clear differences in the function of the culture media used. Interestingly, PCNA immunostaining reaffirmed the high sensibility of EC-derived chondrocytes to the use of CM during MT formation. Actually, these results reveal an evident decrease in cell proliferation due to the use of CM within EC-MTs. These results are in agreement with the morphometric and/or metabolic findings discussed above and also support the hypothesis that the use of CM induces an advanced state of differentiation and/or quiescence on these cells during the MT formation process, with these differences being clearer in EC-derived cells.

Similarly, the differences in the cell behavior and sensibility to the culture media used were also observed in the ECM histochemical and immunohistochemical analyses. The evaluation of the acid proteoglycans by using AB histochemical methods clearly demonstrated the efficacy of the CM to induce the production of these molecules by both cell sources during the MT formation process. Interestingly, the EC-derived chondrocytes showed an earlier proteoglycans synthesis than HC-derived cells, again confirming the fast response of these cells to the CM used. Regarding the analysis of fibrillar collagens conducted with the PS histochemical method, some signs of collagen deposition were observed, but differences among experimental conditions were not clear. Moreover, the immunohistochemical identification of collagens type II and IV again reaffirmed the differences in the function of both cells and culture media used. With respect to collagen type II, the distribution pattern observed in HC-MTs was homogeneous independently of the culture media used. In contrast, the use of CM within EC-MTs efficiently stimulated the synthesis of this cartilage-specific ECM molecule during the period analyzed. In relation to collagen type IV, it is not often evaluated in cartilage TE. Collagen type IV is an essential structural molecule of the pericellular matrix [65], and its identification could help to confirm if this structure is formed during MT formation. Our results show signs of collagen type IV synthesis in the MTs generated during the period studied. In this sense, collagen type IV immunostaining differed in the function of the cell type used, but, surprisingly, no differences were observed in relation to the culture media used to generate the MTs. In this study, collagen type IV was more evident in EC-MTs, where a distinctive pericellular pattern was observed, suggesting the establishment of the pericellular matrix on some MT-forming cells. However, the formation process of the pericellular matrix ex vivo remains poorly understood and more time course and future molecular studies are still needed. Furthermore, orcein staining was negative, confirming that mature elastic fibers were not synthetized within the EC-MTs. Elastic fibers are composed of an elastin-rich core protein (tropoelastin polymerized monomers) and a variable amount of microfibrils, such as fibrillin. Tropoelastin monomers are extracellularly secreted and assembled by polymerization, where diverse ECM components and the tropoelastin monomers concentration play an essential role [66]. These fibers were produced by EC-derived chondrocytes once encapsulated in the nanostructured fibrin–agarose biomaterials [10], suggesting that the scaffold used favored the synthesis of these molecules ex vivo. In this context, our results indicate that EC-MTs did not reach a degree of differentiation in which mature elastic fiber synthesis occurs. These results could be related to the lack of an ECM, which, in this case, was produced de novo by the MT-forming cells, and thus further analyses are needed to determine if these cells are able or not able to produce elastin fiber precursors under these culture conditions.

Finally, the overall morphometric, metabolic and histological results obtained in this study confirm that the use of CM favors the chondrogenic differentiation during the MT formation process, with these results being more evident within the EC-derived chondrocytes. These results could represent a promising alternative to develop new engineered substitutes for cartilage regeneration, supplying viable, functional and biomimetic 3D cellular structures which may contain key molecules to enhance the regenerative potential of cartilage TE treatments. However, it is well known that a full chondrogenic differentiation ex vivo is still difficult to achieve, and thus our results are in accordance with most ex vivo engineered cartilage substitutes [67]. In this regard, more research is still needed in this field. On the one hand, novel biofabrication techniques, such as biomaterial encapsulation, hypoxic culture conditions and/or mechanical stimulation, have shown positive results and should be explored [10,25,68]. On the other hand, the therapeutic efficacy of these bioengineered products must be preclinically evaluated in animal models for cartilage repair and regeneration. Nevertheless, the promising results obtained with the MT technique in this study could be applied in future studies to different biomedical applications such as skin, cornea or nerve tissue engineering.

## 5. Conclusions

This study demonstrated that HC- and EC-derived chondrocytes cultured in agarose microchips were able to form stable, viable, metabolically active and ECM-rich MTs ex vivo. From the morphostructural point of view, more homogeneous, compact and structurally stable MTs were successfully obtained with the use of CM media, confirming the differential biological behavior between HC- and EC-derived cells. Histology demonstrated that MT-forming chondrocytes expressed the S-100 marker over time and were able to synthesize diverse cartilage-specific ECM molecules. The synthesis of the ECM was especially more favorable when the MTs were developed with the use of CM. These findings support the hypothesis that the use of CM provides the necessary factors to favor chondrogenic differentiation, where, in this study, more functional and biomimetic results were obtained between 7 and 14 days of EVD. Interestingly, differences were more remarkable when CM was applied to generate EC-MTs, where cells responded faster and efficiently to this chemical stimulus, obtaining more biomimetic results. These results highlight the importance of culture methods and differences between cell sources employed, presenting the MT culture technique as a promising alternative for its application to a wide range of TE advanced therapies, such as nerve, cornea and/or skin, where the MT technique may contribute to generating more functional and biomimetic 3D substitutes. Finally, future studies are still needed to determine the successfulness and therapeutic efficacy of these MTs in cartilage TE and in vivo preclinical studies.

## Figures and Tables

**Figure 1 biomedicines-09-00292-f001:**
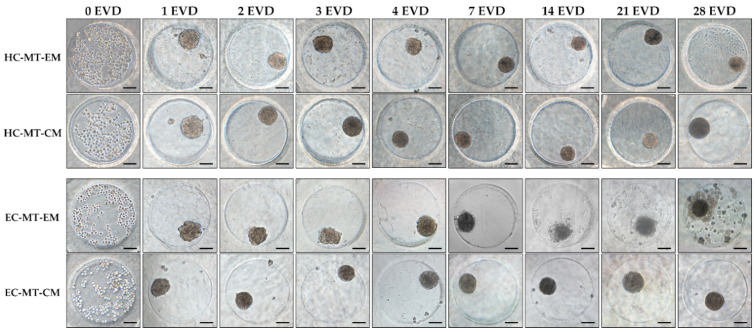
Time course phase-contrast microscopy of human chondrogenic microtissues (MTs) formation process. Note how the cells appear spread within the microchip at 0 days of ex vivo development (EVD), and after 24 h, they associate, forming MTs. Images show hyaline cartilage-derived microtissues (HC-MTs) and elastic cartilage-derived microtissues (EC-MTs) cultured in expansion media (EM) or chondrogenic media (CM). Scale bar = 100 µm.

**Figure 2 biomedicines-09-00292-f002:**
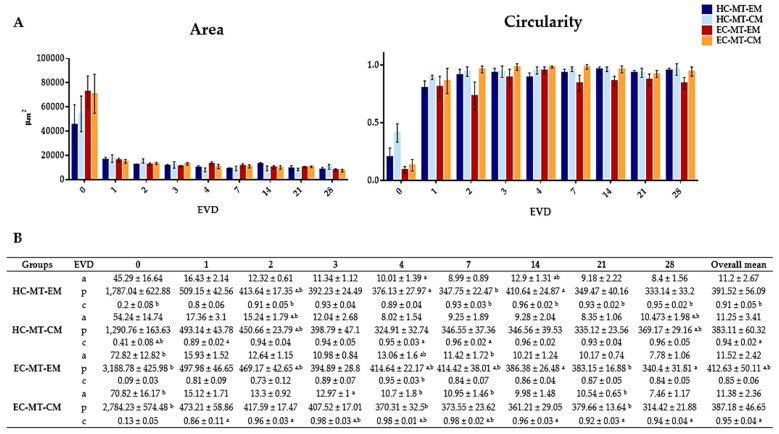
Graphical representation of Area and Circularity of human chondrogenic MTs for each day of EVD (**A**). Quantitative results of morphometric analysis of human chondrogenic MTs (**B**). Results are shown as mean ± SD values for area (**a**) (µm^2^ × 103), perimeter (**p**) (µm^2^) and circularity (**c**) for each experimental group and days of EVD. Here, *p* < 0.05 was considered statistically significant for the Mann–Whitney non-parametric test. Significant differences are indicated as follows: a Differences between MTs (HC or EC) with EM and their respective cell source MT with CM. b Differences between HC-MTs with EM or CM and EC-MTs with the same culture media.

**Figure 3 biomedicines-09-00292-f003:**
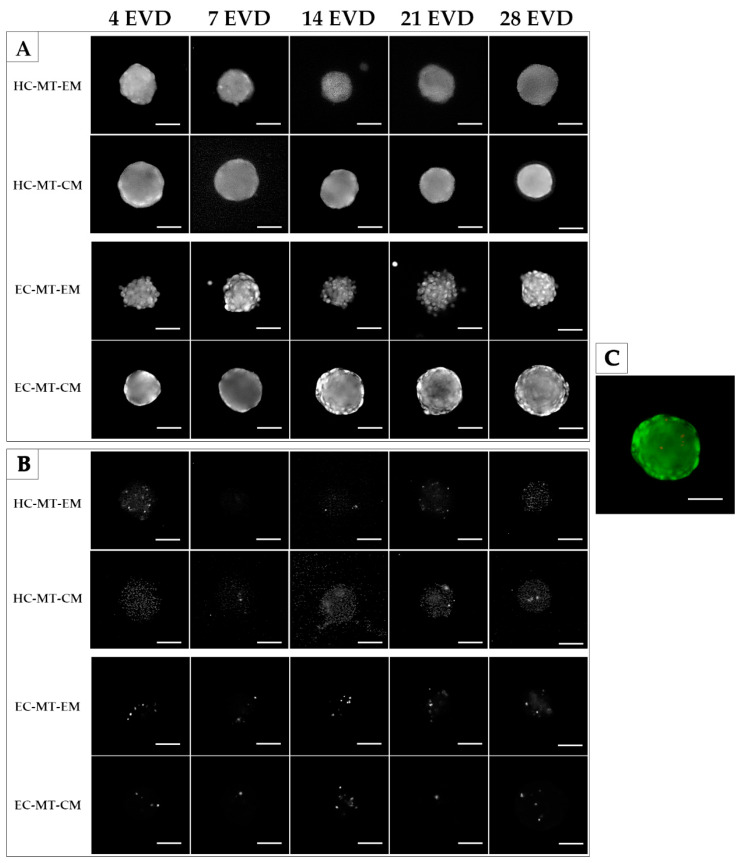
Time course of the Live/Dead assay of human chondrogenic MTs with CM and EM culture conditions. Color live/dead images were split into green channel (**A**) and red channel (**B**). A shows viable cells while B indicates the dead cells for each experimental condition. Note that all MTs generated are mainly composed of viable cells. A representative color image is also provided on the right (**C**). Scale bar = 50 µm.

**Figure 4 biomedicines-09-00292-f004:**
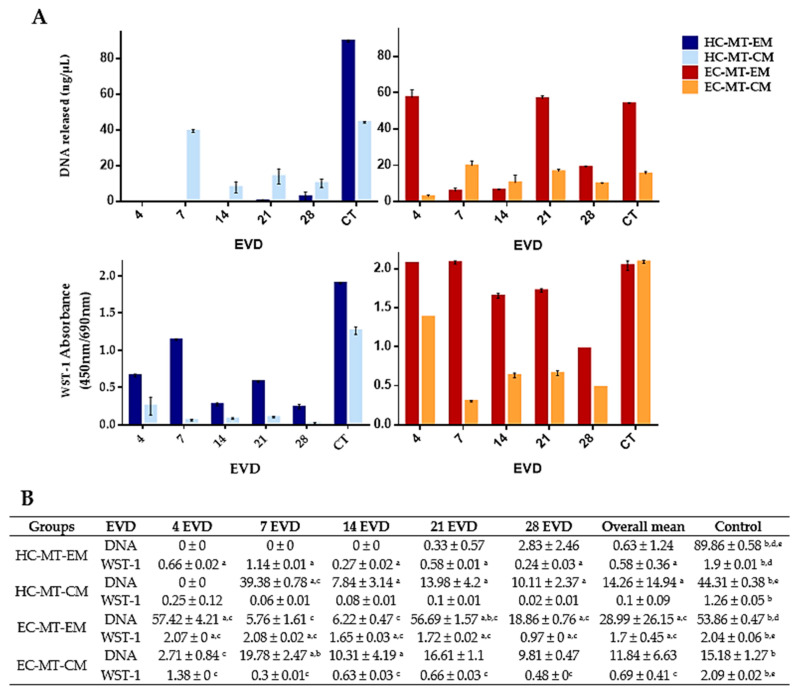
Graphical representation of DNA released (ng/µL) of human chondrogenic MTs for each day of EVD (**A**). Quantitative results of the DNA released and WST-1 biochemical test values of MT-forming human chondrocytes for each day of EVD (**B**). Results are shown as mean ± SD values for DNA in ng/µl and for WST-1 in absorbance (450 nm/690 nm) for each experimental group and day of ex vivo development (EVD). In this study, *p* < 0.05 was considered statistically significant for the Mann–Whitney nonparametric test. Significant differences between groups are indicated with letters as follows: **^a^** Differences between MTs (HC or EC) with EM and their respective cell source MT with CM. ^b^ Differences between MTs (HC or EC with EM or CM) and their respective cell source 2D media control. ^c^ Differences between HC-MTs with EM or CM and EC-MTs with the same culture media. ^d^ Differences between HC- or EC-2D control with EM and their respective cell source 2D control with CM. ^e^ Differences between HC-2D control with EM or CM and EC-2D control with the same culture media.

**Figure 5 biomedicines-09-00292-f005:**
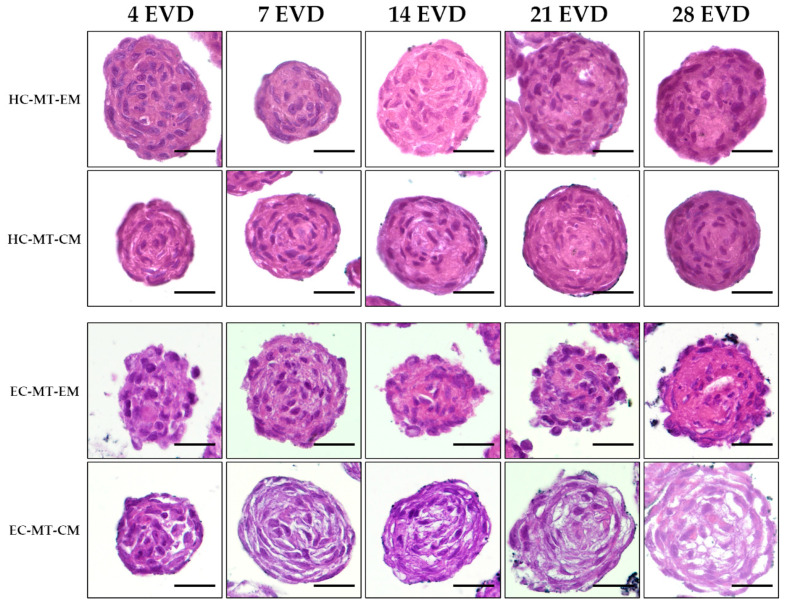
Time course of hematoxylin-eosin (HE) light microscopy analysis of HC-MTs and EC-MTs under CM and EM conditions. HE shows homogenous cell distribution within all chondrogenic MTs. Note the general lower peripheral compaction in the case of EC-MT-EM with respect to the other conditions. For all pictures, scale bar = 50 µm.

**Figure 6 biomedicines-09-00292-f006:**
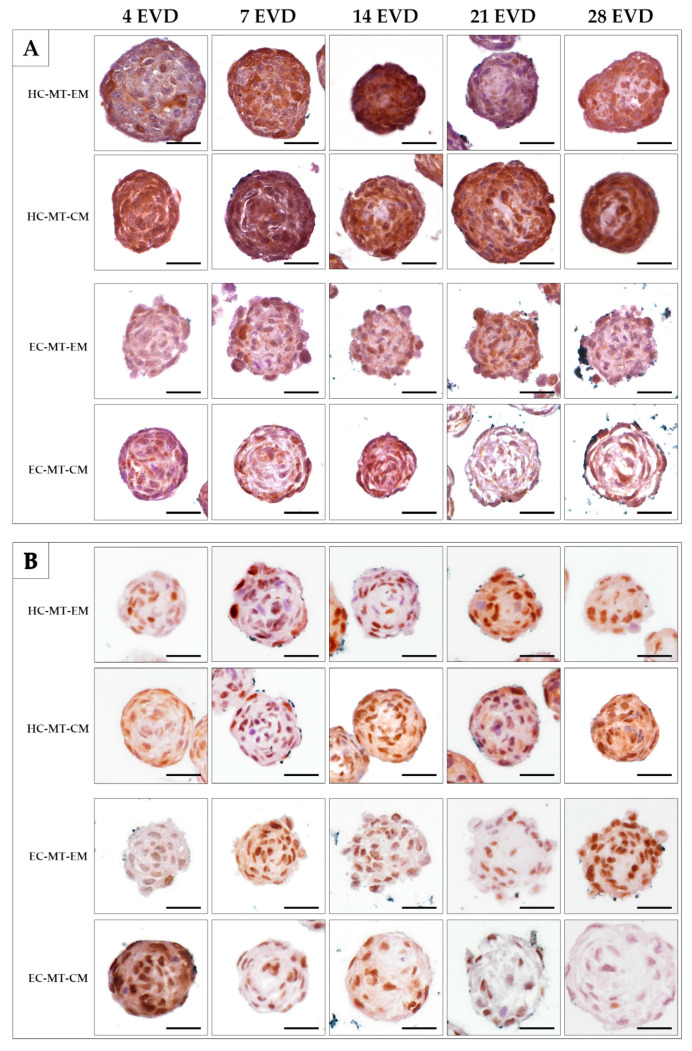
Time course of histochemical analysis of chondrogenic marker S-100 (**A**) and proliferating marker PCNA (**B**) in the HC-MTs and EC-MTs cultured in EM and CM. Note that chondrogenic MTs show an S-100-positive reaction in all experimental conditions, being higher in HC-MTs (**A**). It is possible to observe that MTs are composed of abundant proliferating cells, where, in the case of EC-MTs, differences were observed with the use of CM (**B**). Scale bar = 50 µm.

**Figure 7 biomedicines-09-00292-f007:**
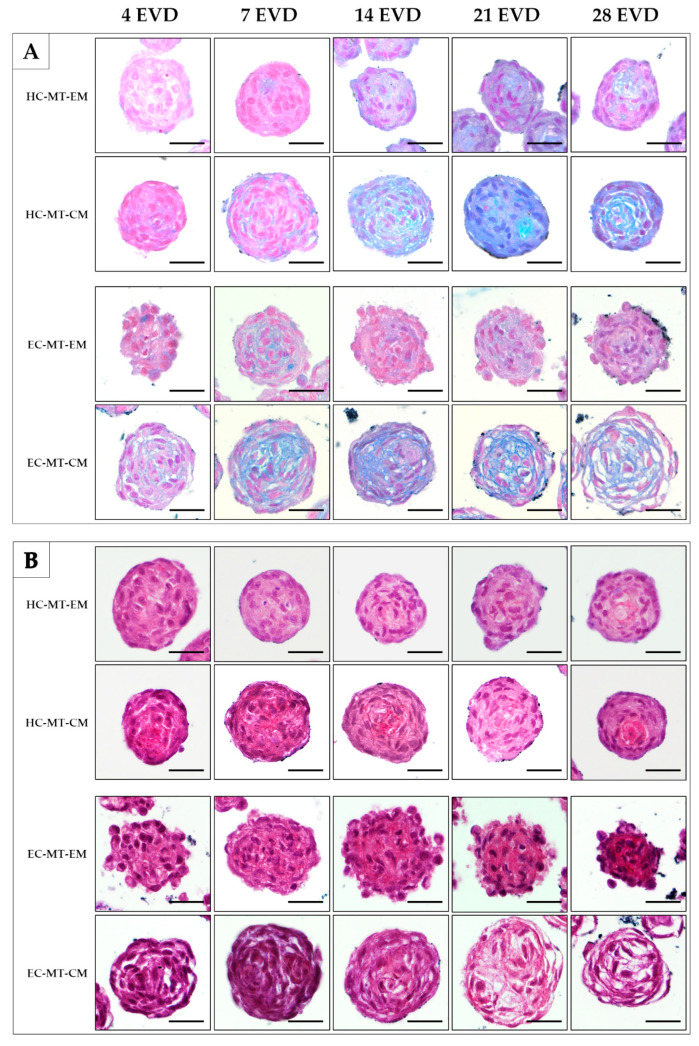
Histological and histochemical analyses of HC-MTs and EC-MTs under CM and EM. Alcian blue (AB) stains acid proteoglycans in light blue (**A**). Picrosirius (PS) stains the fibrillar collagen fibers in red (**B**). Note that CM-MTs show higher AB staining than EM-MTs (**A**). For all pictures, scale bar = 50 µm.

**Figure 8 biomedicines-09-00292-f008:**
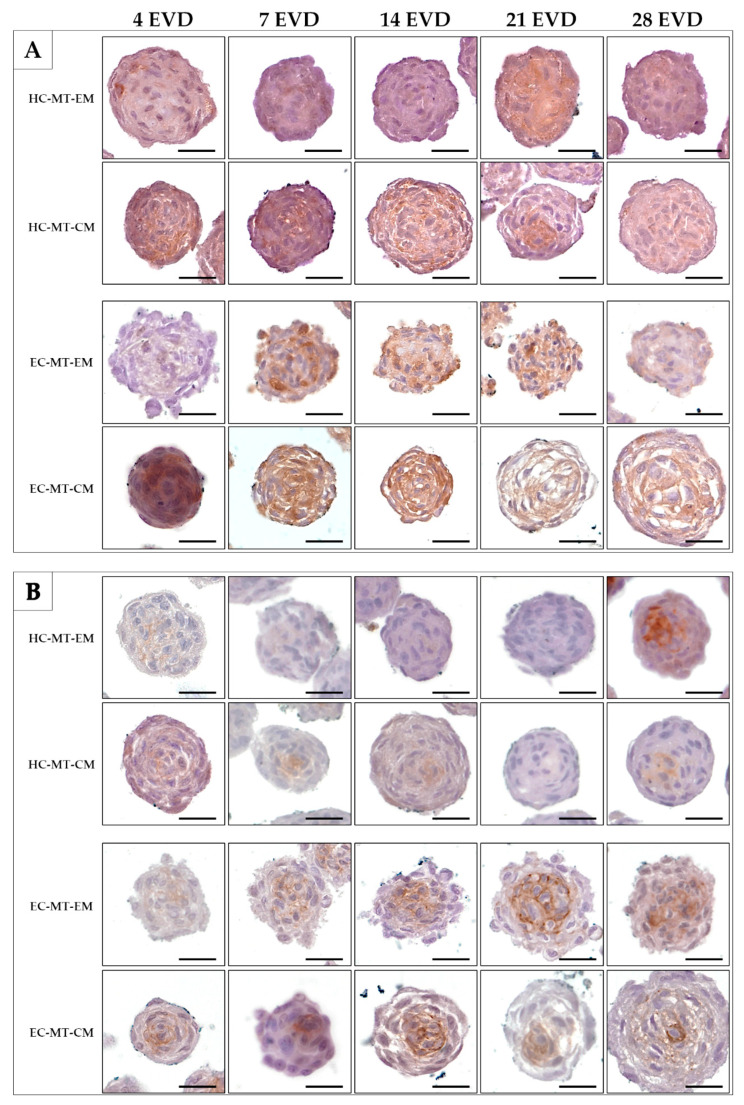
Histochemical analysis of expression of collagen type II (Col II) (**A**) and collagen type IV (Col IV) (**B**) of HC-MTs and EC-MTs under CM and EM. CM-MTs show higher reaction to Col II (**A**) in both cell types. Note the positive reaction to Col IV in EC-MTs (**B**). For all pictures, scale bar = 50 µm.

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
