# Peer review of "Ex Vivo Generation and Characterization of Human Hyaline and Elastic Cartilaginous Microtissues for Tissue Engineering Applications"

_biomedicines, 2021, doi:10.3390/biomedicines9030292_

Round 1

Reviewer 1 Report

In this paper, a method was proposed to make a cartilaginous micro-tissue.  I think that the reproducibility is important, but it was not confirmed. I think that the number of trials should be written. Also, I think that the quantitive evaluation for differentiation, such as gene expression and image analysis of immunestained MTs,  is needed. 

Author Response

R/. Dear reviewer, thank you very much for your time and valuable comments regarding our manuscript. Concerning reproducibility, all tests were conducted at least in triplicate and this information is available for readers. Moreover, several tests and optimizations were done before the definitive experimental set-up included within this article. In relation to the quantitative analysis of the differentiation process. We are planning to include quantitative genetic evaluation in future biomaterial-based and in vivo studies. Finally, the English was corrected in the revised version of the manuscript.

Reviewer 2 Report

The manuscript titled "Ex Vivo Generation and Characterization of Human Hyaline and Elastic Cartilaginous Microtissues for Tissue Engineering Applications" describes the formation of hyaline and elastic cartilage microtissues with microplates and observed cartilage tissue formation with proliferating cells, cartilage histological staining. 
The result shows a faster and a more sensitive response of the elastic cartilage derived cells to the use of chondrogenic medium than hyaline chondrocytes. 
This manuscript has reasonable approach to compare hyaline cartilage and elastic cartilage-derived chondrocytes application to cartilage tissue regeneration, finding effective conditions for EC and HC tissue formation. 
This manuscript is well written and has little points to be addressed and ready to be published as it is. 

Minor points;
page 3 line 105 mercure=> mercury
page 10 Fig upper right legend is partly shown, need to be corrected. 

Author Response

R/. Dear reviewer, thank you very much for your positive and valuable comments regarding our manuscript. It was quite difficult to organize all these experiments together and construct a unique message concerning the potential usefulness of these novel chondrogenic-based microtissues. We would like to thank you for the corrections made to our manuscript. Furthermore, the final version of the manuscript was revised by a professional on language and proofreading service.

Minor points;
page 3 line 105 mercure=> mercury

R/.The word was corrected accordingly

page 10 Fig upper right legend is partly shown, needs to be corrected.

R/.Apologize for this format error. It was corrected in the revised version of our manuscript.